# Protein Kinase RhCIPK6 Promotes Petal Senescence in Response to Ethylene in Rose (*Rosa Hybrida*)

**DOI:** 10.3390/genes13111989

**Published:** 2022-10-31

**Authors:** Yanqing Wu, Lanxin Zuo, Yanxing Ma, Yunhe Jiang, Junping Gao, Jun Tao, Changxi Chen

**Affiliations:** 1Joint International Research Laboratory of Agriculture and Agri-Product Safety, The Ministry of Education of China, Institutes of Agricultural Science and Technology Development, Yangzhou University, Yangzhou 225009, China; 2Beijing Key Laboratory of Development and Quality Control of Ornamental Crops, Department of Ornamental Horticulture, College of Horticulture, China Agricultural University, Beijing 100193, China; 3College of Horticulture and Landscape Architecture, Yangzhou University, Yangzhou 225009, China

**Keywords:** rose, flower senescence, ethylene, protein kinase, CIPK6

## Abstract

Cultivated roses have the largest global market share among ornamental crops. Postharvest release of ethylene is the main cause of accelerated senescence and decline in rose flower quality. To understand the molecular mechanism of ethylene-induced rose petal senescence, we analyzed the transcriptome of rose petals during natural senescence as well as with ethylene treatment. A large number of differentially expressed genes (DEGs) were observed between developmental senescence and the ethylene-induced process. We identified 1207 upregulated genes in the ethylene-induced senescence process, including 82 transcription factors and 48 protein kinases. Gene Ontology enrichment analysis showed that ethylene-induced senescence was closely related to stress, dehydration, and redox reactions. We identified a calcineurin B-like protein (CBL) interacting protein kinase (CIPK) family gene in *Rosa hybrida*, *RhCIPK6*, that was regulated by age and ethylene induction. Reducing *RhCIPK6* expression through virus-induced gene silencing significantly delayed petal senescence, indicating that RhCIPK6 mediates petal senescence. In the *RhCIPK6*-silenced petals, several senescence associated genes (*SAGs*) and transcription factor genes were downregulated compared with controls. We also determined that RhCIPK6 directly binds calcineurin B-like protein 3 (RhCBL3). Our work thus offers new insights into the function of CIPKs in petal senescence and provides a genetic resource for extending rose vase life.

## 1. Introduction

Flowers are highly evolved organs that help angiosperms achieve reproductive success [1]. Although petals are not directly involved in reproduction, they function to attract potential pollinators [2]. In nature, petals senesce and shed after pollination or following flower opening. Petal senescence is characterized by rolling, wilting, fading, and, eventually, programmed cell death and abscission [3,4]. For the floral industry, increasing petal longevity is crucial to maximize the vase life of flowers.

Petal senescence is affected by multiple autonomous and environmental factors [4,5], among which phytohormones are the most important internal factors. Ethylene, abscisic acid (ABA), and jasmonic acid induce petal senescence, whereas cytokinin, gibberellin, and salicylic acid inhibit it [6,7,8,9]. Ethylene, in particular, broadly functions as an endogenous regulator of leaf and flower senescence and fruit ripening [10]. In ethylene-sensitive plants, exogenous ethylene treatment accelerates ethylene production and hastens petal senescence, whereas inhibitors of ethylene biosynthesis or signaling delay petal senescence [11]. Almost all cut rose cultivars are sensitive to ethylene [3]. Ethylene induces the ethylene receptor gene *RhETR3* and its downstream signal transduction components *RhCTR1/2* to promote petal senescence [3,12,13]. Senescence-associated genes (SAGs) are often used as molecular markers of senescence in plants [14,15]. SAGs can be induced in age and developmental progress and by application of exogenous hormones [15]. In petunia, the ectopic expression of the *SAG12*pro: *IPT* gene delays flower senescence [16]. Recent studies have revealed that ethylene modulates flower senescence by regulating the expression of some senescence-associated genes in rose. Several transcription factors including RhHB1/6, RhMYB108, RhERF113, and RhWRKY33 were reported to regulate ethylene-induced petal senescence and the expression of *RhSAG12* (the rose homolog of *SAG12*) [6,7,17,18]. However, the molecular mechanism of ethylene-induced flower senescence remains largely unknown.

Protein kinases regulate plant growth, development, and stress responses, and reversible protein phosphorylation is a key regulatory factor in many cellular signaling networks [19]. Recent reports of protein kinases related to leaf senescence emphasize the importance of phosphorylation in this process as well. For example, the *Oryza sativa* S-domain receptor-like kinase OsSIK2 negatively regulates the leaf senescence induced by darkness and abiotic stress [20]. Mitogen-activated protein kinases (MAPKs) are associated with hormone-induced leaf senescence [21,22]. SNF1-RELATED KINASE1 (SnRK1) directly phosphorylates ETHYLENE INSENSITIVE3 (EIN3) to delay ethylene-induced leaf senescence. Additionally, some protein kinases are involved in signal transduction, such as calcium-dependent protein kinases (CDPKs) and CBL-interacting protein kinases (CIPKs), which participate in stress signal transduction in plants [23]. Although protein kinases represent one of the largest gene families in eukaryotes, the functions of most plant kinases have not yet been reported [24], and only a few have been identified for their role in flower senescence. Characterization of novel kinases could help reveal the underlying mechanisms of flower senescence.

The objective of this study was to identify novel regulators mediating ethylene-induced senescence in rose petals. Transcriptional regulators and protein kinases including the CIPK family protein kinase, RhCIPK6, were identified by RNA sequencing. We characterized the RhCIPK6 function through virus-induced gene silencing (VIGS) and identified several downstream genes in the signal cascade. To the best of our knowledge, this is the first demonstration of CIPK involvement in the ethylene-induced senescence response in rose. Our results thus provide new insights of the regulatory mechanism of ethylene-regulated petal senescence.

## 2. Materials and Methods

### 2.1. Plant Materials and Treatments

Rose (*R. hybrida* ‘Samantha’, Rh) plants were grown in a greenhouse. The harvest and pretreatment procedures of cut rose flowers were previously described [25]. Rose plantlets were grown in a climate chamber (22 ± 1 °C, 60% relative humidity, and light–dark photoperiods of 16/8 h). Rose stems were rooted in a 1:1 mixture of peat and vermiculite for about 4 weeks for the silencing assay.

Petal samples at stages 1–6 were harvested from the middle whorl of flowers after ethylene treatment. Flowers at stage 2 were used for hormone treatments. Flowers in vases were sealed in an airtight chamber exposed to 10 μL/L ethylene for 24 h, whereas untreated flowers were sealed with air. For other hormone treatments, flowers were placed in vases containing 100 μmol/L abscisic acid (ABA), 100 μmol/L methyl jasmonate, 100 μmol/L naphthylacetic acid (NAA), and 100 μmol/L 6-benzylaminopurine (6-BA) for 24 h. Mock samples were placed in water with the corresponding solvent.

### 2.2. RNA-Seq Analysis and Annotation

Petals at different developmental stages and after ethylene treatment and control flowers were collected. The samples were sent to LC Biotech Corporation (Hangzhou, China) for RNA-isolation and RNA sequencing. RNA libraries were sequenced using the Illumina HiSeq2500 system (Illumina, San Diego, CA, USA). Read processing and unigene assembly were performed as previously described [26]. RNA-seq reads were examined to remove low-quality (Q value < 20) reads and adapters using Trimmomatic [27]. High-quality clean reads were assembled de novo into contigs using the Trinity program [28]. To remove the redundancy of Trinity-assembled contigs, the contigs were further assembled de novo using iAssembler [28]. Genes were annotated based on the reference *Rosa chinensis* ‘Old Blush’ genome database [29]. The unigenes were compared with protein database sequences using BLASTX [30]. Using the FPKM method (fragments per kilobase of transcript sequence per million base pairs sequenced) [31], a differential expression analysis of four groups of genes was performed using DESeq2 [32]. Gene Ontology (GO) terms were assigned to the rose assembled transcripts based on the GO terms annotated to their corresponding homologues in the UniProt database. Biochemical pathways were predicted from the rose transcripts using Pathway Tools [33].

### 2.3. RNA Extraction and Quantitative Reverse-Transcription PCR (qRT-PCR)

Total RNA was extracted from rose petals using the hot borate method as previously described [12]. The cDNA templates were synthesized from 1 μg total RNA using the HiScript^®^ II reverse transcriptase kit (Vazyme, Nanjing, China) according to the manufacturer’s instructions. qRT-PCR was performed in a Step One Plus real-time PCR system (Applied Biosystems, Carlsbad, CA, USA) using an M5 HiPer real-time PCR super mix (Mei5 Biotech, Beijing, China). The ubiquitin *RhUBI2* gene was used as an internal control [34].

### 2.4. Cloning and Sequence Analysis

The open-reading frames of *RhCIPK6*, *RhCBL1*, and *RhCBL3* were amplified from *R. hybrida* ‘Samantha’ petal cDNA templates. Primers were designed based on predicted sequences in the *R. chinensis* database (https://lipm-browsers.toulouse.inra.fr/pub/RchiOBHm-V2/; accessed on 15 March 2020) and rose transcriptome database (http://bioinfo.bti.cornell.edu/cgi-bin/rose_454/index.cgi; accessed on 15 March 2020). Putative CIPK protein sequences from other species were obtained from the *Arabidopsis* genome database (https://www.arabidopsis.org; accessed on 20 May 2020) and NCBI database (https://blast.ncbi.nlm.nih.gov; accessed on 20 May 2020). The CIPK family phylogenetic tree was constructed using the maximum likelihood method in MEGA7.0 [35]. The sequence alignment of RhCIPK6 with its homologues from other plants was performed through a BLAST search (https://blast.ncbi.nlm.nih.gov/Blast.cgi; accessed on 20 May 2020). PCR primers are listed in Appendix A.

### 2.5. Virus-Induced Gene Silencing

The virus-induced gene silencing assay was performed as previously described [36]. An *RhCIPK6*-specific fragment was inserted into the TRV2 vector [37]. The constructs pTRV1, pTRV2, and pTRV2-*RhCIPK6* were transformed into *Agrobacterium tumefaciens* GV3101. Mixed cultures containing pTRV1 or pTRV2-*RhCIPK6* in a 1:1 ratio were used as a reporter, and *A. tumefaciens* GV3101/pTRV1 and *A. tumefaciens* GV3101/pTRV2 were used as controls. Rose plantlets were immersed in the above cultures, and the Agrobacteria were infiltrated into seedlings by the vacuum method. The plantlets were grown in a growth chamber (22 ± 1 °C, 60% relative humidity, and light–dark photoperiods of 16/8 h). PCR primers are listed in Appendix A.

### 2.6. Protein–Protein Interaction Network

The STRING v11.5 database (http://string-db.org; accessed on 2 June 2021) was used to construct the protein–protein interaction network of RhCIPK6 [38]. Only interactions with a score ≥ 0.7 are shown. Evidence of interaction is the distance between nodes and closer nodes having a higher score. Non-interacting proteins were not shown.

### 2.7. Yeast Two-Hybrid Assays

A yeast two-hybrid (Y2H) system was used to screen for RhCIPK6-interacting proteins [39]. The coding sequence of RhCIPK6 was inserted into the bait vector pGBKT7. The coding sequences of RhCBL1 and RhCBL3 were inserted into the prey vector pGADT7. The empty vectors were used as negative controls. The bait and prey vectors were co-transformed into the *Saccharomyces cerevisiae* strain Y2H Gold. The transformants were then spotted onto Synthetic Defined media/-Trp-Leu, SD/-Trp-Leu-His + 5 mM 3-amino-1, 2, 4-triazole (3-AT). PCR primers are listed in Appendix A.

### 2.8. Split-luciferase Complementation Assays

The split-luciferase complementation assay was performed as previously described [40]. The CDS of RhCIPK6 and RhCBL3 were amplified and cloned into pCAMBIA1300-Cluc or pCAMBIA1300-Nluc, respectively. RhCBL3-Nluc or Nluc was co-transformed with Cluc-RhCIPK6 or Cluc into *Nicotiana benthamiana* leaves via *A. tumefaciens*. After 3 days, 100 μM d-luciferin was sprayed on the leaves and luminescence intensities were detected using a Nightshade LB985 CCD camera (Berthold, Germany). PCR primers are listed in Appendix A.

## 3. Results

### 3.1. Ethylene Accelerates Rose Petal Senescence

To investigate the effect of ethylene on gene expression during flower senescence, we compared the expression of the senescence marker gene *RhSAG12* obtained by qRT-PCR during developing petal senescence and upon ethylene induction. The flower development process was divided into six stages [3] (Figure 1A). Before stage 3, *RhSAG12* expression was hardly detected in the petals; however, the transcript abundance of *RhSAG12* increased by nearly 100-fold from stages 3 to 5 (Figure 1C). When flowers at stage 2 were treated with ethylene, the rose flowers showed greater petal opening angle and petal curling [25] (Figure 1B). Compared with the control petals, the expression of *RhSAG12* in stage 2 was considerably induced by ethylene after 24 h of treatment (Figure 1D). Our results suggest that ethylene promotes the initiation of petal senescence, and changes in gene expression for senescence precede overt phenotypic changes.

### 3.2. Identification of Unique DEGs Associated with Ethylene-Related Senescence in Rose Petals

To determine gene alterations during the onset of ethylene-induced petal senescence, RNA-seq libraries were prepared from rose petals at different stages (stage 3 and 5) as well as with and without ethylene treatment (stage 2), respectively. We performed analyses on the developmental data at stage 5 versus stage 3. We found that among the 7992 differentially expressed genes (DEGs) that had detectable expression in petals, 4091 genes were upregulated and 3901 were downregulated compared with the expression levels at stage 3 (Figure 2A and Appendix A). We also constructed a transcriptome database of ethylene-treated rose petals. A total of 4044 DEGs showed dramatic changes upon ethylene treatment. Among them, 2330 genes were upregulated, whereas 1714 genes were downregulated (Figure 2A and Appendix A). We compared the upregulated DEGs in the developmental-dependent process with those in the ethylene-induced process (Figure 2B). The two groups shared 1207 DEGs, which were the candidates that may be involved in ethylene-induced senescence (Figure 2B and Appendix A).

We carried out gene ontology (GO) analysis on these 1207 DEGs to evaluate their potential functions. In the biological process category, the subcategories of “translation”, “regulation of transcription”, “oxidation–reduction process”, and “protein phosphorylation” were dominant (Figure 3A). This suggests that numerous metabolic changes occur during petal senescence. In addition, several GO terms were related to water deprivation and abscisic acid, indicating that ethylene-induced senescence is closely related to the water balance of petals (Figure 3A). In the molecular function category, the abundant groups included ATP binding, protein binding, and zinc ion binding (Figure 3B). Other appealing groups included nucleic acid binding transcription factor activity and protein serine/threonine kinase activity (Figure 3B).

Transcription factors (TFs) and protein kinases are crucial signaling regulators that play important roles in several plant developmental processes and stress responses. Among the 1207 DEGs, we identified 82 that encode TFs and 48 that encode protein kinases (Appendix A). Among those we identified, some TF genes have similarities to known genes, such as *ANAC092* and *WRKY40*, which are linked to senescence and autophagy in *Arabidopsis* [41,42]. In addition, senescence-related TF genes, including *RhWRKY33* and *RhHB6*, have been found in rose [8,18]. As for protein kinases, except for a few reports on the regulation of mitogen-activated protein kinases, there is not a lot of amount of data about protein kinases involved in rose plant growth and development [43]. Therefore, we chose protein kinases for further study.

### 3.3. RhCIPK6 Transcription Is Induced during Petal Senescence

Calcineurin B-like (CBL) interacting protein kinases (CIPKs) have been reported to be widely involved in various stress responses, hormone signal transduction, and development regulation [44]. We found a CIPK family gene *RchioBHmChr2g0085831* among the 48 candidate protein kinases. To identify *CIPK* genes in the rose genome, we performed a BLAST search using known *Arabidopsis* CIPK protein sequences. In the rose ‘Old Blush’ genome, we found 20 candidate CIPK proteins. We investigated the 20 *CIPKs* in our rose transcriptome database (http://bioinfo.bti.cornell.edu/cgi-bin/rose_454/index.cgi) and found that 15 of the 20 *CIPK* genes were detected in petals, with *RchioBHmChr2g0085831* displaying the highest transcript accumulation during petal opening. Compared with 26 CIPKs in *Arabidopsis*, rose has fewer CIPK genes. To understand the evolutionary relationship of protein kinases in rose, a phylogenetic tree was constructed with full-length CIPK protein sequences from rose and *Arabidopsis* (Figure 4A).

Phylogenetic tree analysis showed that the protein encoded by *RchioBHmChr2g0085831* had high sequence homology with *AtCIPK6*, so this gene was renamed *RhCIPK6* (Figure 4A). RhCIPK6 encodes a putative protein of 428 amino acids. Compared with CIPK6s from different species, RhCIPK6 is predicted to contain a typical kinase domain (catalytic domain) and a regulatory domain (calmodulin-like domain) (Figure 4B). In addition, the CIPK6 protein showed high similarity to those of CIPK6 genes of *Fragaria vesca* (Fv), *Prunus armeniaca* (Pa), *Prunus persica* (Pp), *Malus domestica* (Md), and *Arabidopsis thaliana* (At) with identities of 93, 91, 91, 90, and 78%, respectively (Figure 4B). These results indicate that CIPK6 proteins are evolutionarily conserved across plant species.

The qRT-PCR expression analysis showed that the transcriptional abundance of *RhCIPK6* in petals increased during flower opening and peaked in senescent flowers (Figure 5A). When the petals were treated with hormones, we observed that exogenous ethylene and abscisic acid (ABA) significantly induced *RhCIPK6* expression, whereas jasmonic acid, naphthylacetic acid, and cytokinin treatments did not alter *RhCIPK6* transcript levels (Figure 5B).

### 3.4. Silencing of RhCIPK6 Delays Rose Flower Senescence

To investigate the potential role of RhCIPK6 in flower senescence, *RhCIPK6* was silenced using virus-induced gene silencing (VIGS). We selected a 464 bp fragment from an *RhCIPK6*-specific region to silence the *RhCIPK6* expression. The qRT-PCR results showed that *RhCIPK6* transcriptional abundance was significantly reduced in *RhCIPK6*-silenced (pTRV2-*RhCIPK6*) petals compared with empty vector controls (Figure 5C). However, in the ornamental value phases, *RhCIPK6* silencing caused a delayed petal senescence phenotype, with stages 5 and 6 lasting 6.5 ± 0.2 days compared with 4.7 ± 0.3 days in the empty vector control (Figure 5D,E).

We performed a large-scale screening of DEGs in *RhCIPK6*-silenced plants compared with empty vector control plants using RNA-seq. In *RhCIPK6*-silenced plants, we found 1077 DEGs. Comparison of these DEGs with those involved in developmental and ethylene-induced senescence revealed 224 positively correlated DEGs: 23 upregulated DEGs and 201 downregulated DEGs (Appendix A). The GO analysis of these 224 DEGs showed that in the biological process category, terms such as “response to abscisic acid” and “response to water deprivation” were highly represented, which is consistent with the GO terms enriched in ethylene-related senescence (Figure 6A). Additionally, several GO terms related to cell-wall biogenesis were also enriched, indicating that RhCIPK6 may play a role in cell-wall reorganization. Some senescence- and stress-associated genes previously reported, including those encoding SAGs and TFs, were significantly downregulated in *RhCIPK6*-silenced petals, which is consistent with the senescence phenotype (Figure 6B).

### 3.5. RhCIPK6 Interacts with RhCBL3

To explore potential interacting proteins of RhCIPK6, we constructed a protein–protein interaction network using the STRING database. RhCIPK6 potentially interacts with four CBLs, two potassium channels, two protein phosphatases, and two DNA polymerases (Figure 7A and Appendix A). CBLs specifically target CIPKs to form a signaling network [44]. Among the four CBL protein genes, *RchiOBHm_Chr7g0205861* (*RhCBL1*) and *RchiOBHm_Chr5g0075551* (*RhCBL3*) transcripts were detected in petals, where *RhCBL3* showed greater transcript accumulation (Appendix A). To verify the interaction between RhCIPK6 and 2 of the identified CBLs, we performed a yeast two-hybrid analysis using RhCIPK6 fused to the GAL4 DNA-binding domain as the bait and either of the two CBL proteins fused to the GAL4 activation domain as the prey. Compared with the controls, yeast coexpressing RhCIPK6 and RhCBL3 normally grew on selection medium, indicating that RhCIPK6 physically interacts with RhCBL3, but not with RhCBL1 (Figure 7B). To validate this interaction, we performed a split-luciferase complementation (Split-LUC) assay in *N. benthamiana* leaves. As shown in Figure 7C, luciferase activity was reconstituted when Cluc-RhCIPK6 and Nluc-RhCBL3 were coexpressed, whereas no LUC activity was detected in the control. These results demonstrate that RhCIPK6 directly interacts with RhCBL3.

## 4. Discussion

Floral longevity is an important functional and commercial quality trait for ornamental plants. The physiology and molecular mechanisms of flower senescence need to be understand to maximize floral longevity. In most cases, flower senescence often refers to the petal senescence. Petal senescence is a precisely controlled and irreversible process [1]. This process involves many morphological, physiological, cytological, and molecular changes. Petal senescence is the final stage of petal development, typically characterized by color change and wilting (Figure 1) [5]. Genetic and anatomical changes associated with petal senescence take place before visible morphological changes [45]. During petal senescence, color change is tightly linked to the variation in the pigment ratio [46,47]. In rose petals, high pH alters the structure of anthocyanin pigments, resulting in a blue color [48]. Several other observations have been implicated in this process, including water loss, ion leakage, reactive oxygen species (ROS) production, the decline of membrane integrity, and the breakdown of macromolecules and organelles in senescent petals. In our transcriptome and proteome, proteins associated with hydrolase activity were significantly enriched during developmental senescence (Figure 3) [49], which is consistent with previous studies showing that senescence is accompanied by increased protein proteolysis in petunia and iris [50,51].

Phytohormones are one of the most important regulators in controlling senescence [5]. Studies on flowers have confirmed the important role of phytohormones in regulating organ senescence [1]. Despite numerous studies, the molecular mechanisms underscoring the relationship between phytohormones and senescence have not been fully elucidated. The effect of exogenous ethylene on rose flowers was rapid and dramatic, especially the expression of the senescence marker gene (Figure 1). Interestedly, ABA response genes were enriched during ethylene-induced senescence (Figure 3). ABA is known to be a positive regulator of leaf senescence, similar to ethylene [5]. In flowers insensitive to ethylene, ABA may be the main phytohormone to accelerate senescence, but the mechanism remains unclear [1]. In flowers insensitive to ethylene, ABA may affect petal senescence in an ethylene-dependent manner [1]. Exogenous ethylene and ABA treatments have been shown to accelerate petal senescence of cut rose [52]. Ethylene treatment increases endogenous ABA levels [52,53], whereas the ethylene inhibitor impedes ABA-induced flower senescence [54]. These observations suggest the crosstalk between ethylene and ABA pathways during flower senescence. In this work, ethylene-related senescence genes were enriched in a number of dehydration-related GO terms, including “response to water deprivation”, “response to abscisic acid”, and “response to osmotic stress” (Figure 3A), indicating that ethylene is potentially involved in wilting during petal senescence. Our previous study showed that the phenotype of rose flowers subjected to dehydration was similar to those that were treated with ethylene [55]. Ethylene as a plant hormone involved in dehydration stress signaling has been confirmed in the flower opening process of rose [55]. Aquaporins are the major channels for water transport across the plasma membrane, and the expression of the functional aquaporin RhPIP2;1 is significantly downregulated by ethylene during petal expansion [56,57]. However, it is still unclear how ethylene signaling affects water loss during petal senescence.

Petal senescence is finely regulated at the transcriptional, post-transcriptional, and post-translational levels. Transcriptional regulation related to petal senescence has been studied in numerous ornamental plants [58,59,60,61], but little is known about the post-translational regulation of this process. Post-translational modifications of proteins, such as phosphorylation, ubiquitination, acetylation, and methylation, are critical for the rapid modulation of signaling events [62]. Protein phosphorylation and dephosphorylation by kinases and phosphatases are two of the most well-studied post-translational modification pathways in leaf senescence [63]. Here, we focused on the protein kinases in this study. We have reported 48 protein kinases that are potentially involved in ethylene-regulated petal senescence, none of which have been studied in rose. Elucidating the signaling pathway of kinases during petal senescence is an important research direction.

Our work here demonstrated that RhCIPK6 is an important regulator during petal senescence (Figure 5). CIPKs are regulated by many factors, including stress, phytohormones, and nutrient deficiencies [64]. The expression pattern of *RhCIPK6* was obviously induced by ethylene treatment (Figure 5B). In carnation petals, the transcript level of a putative *CIPK6* was reported to be upregulated after ethylene treatment [60]. Previous studies have demonstrated that CIPK6 orthologs from different plant species participate in various signaling pathways, such as salt stress, osmotic stress, innate immunity, root development, and sugar homeostasis [65,66,67,68,69]. Here, we showed that RhCIPK6 is involved in the regulation of ethylene-mediated petal senescence and that silencing *RhCIPK6* affected the expression of senescence-related genes (Figure 5 and Figure 6). GhCIPK6a is involved in ROS scavenging in cotton, whereas SlCIPK6 mediates ROS generation in tomato [70,71]. SlCIPK6, AtCIPK11, and AtCIPK26 activate NADPH oxidase (RBOHB/F) and ROS production [68,72]. ROS play a key role in the regulation of senescence and act as a signal for cell death during senescence [73]. It is tempting to speculate that RhCIPK6 might regulate ROS generation during petal senescence, which requires further study. Moreover, we found the enrichment of GO terms linked to cell-wall biogenesis and metabolism in *RhCIPK6*-silenced petals (Figure 6). Previous studies showed that many cell-wall-degrading genes were regulated during senescence [15,63]. Petal abscission is also associated with pectin solubility in the cell wall [74]. Cell-wall degradation generally accompanied programmed cell death during petal senescence [75]. These findings indicate that RhCIPK6 may somehow affect the changes in the cell wall, thereby regulating petal senescence.

In the PPI network, CBLs and potassium channels were predicted to interact with RhCIPK6 (Figure 7). The CBL–CIPK modules are conserved components decoding Ca^2+^ signals in multiple signal transduction pathways [76], and CBLs and CIPKs can form multiple interaction modules [77]. However, the role of Ca^2+^ signal during petal senescence is not completely defined. Here, we showed that RhCIPK6 interacts with RhCBL3 (Figure 7), but it is unclear whether RhCBL3 is involved in ethylene-induced petal senescence. There are potentially other CBL proteins that participate in RhCIPK6-regulated senescence. The protein interaction network showed that RhCIPK6 may regulate the potassium channel AKT1. Potassium is an important macronutrient required for plant growth, and it is implicated in senescence and programed cell death [78,79]. Changes in potassium concentration may affect the turgor pressure and pH of petal cells, and thereby influence fading and wilting of flowers. RhCIPK6 directly interacted with RhCBL3 (Figure 7). AtCBL3 controls ion homeostasis through vacuolar H^+^-ATPase (V-ATPase) in *Arabidopsis* [80]. In morning glory, V-ATPase activity is strictly controlled to regulate vacuolar pH and membrane potential values, which is related to the color change of epidermal cells. Studies to determine the mechanism of RhCIPK6 and RhCBL3 for petal senescence are in progress.

## 5. Conclusions

Our RNA-Seq analysis has permitted us to dissect the transcriptome during ethylene-induced senescence. Among the related candidate genes, we identified the protein kinase RhCIPK6, which is closely related to AtCIPK6 from *Arabidopsis thaliana*. Furthermore, our data demonstrated that RhCIPK6 played an important role in rose petal senescence. This study provided foundations for further research on the regulatory mechanism of protein kinase during petal senescence and gene resources for improving the longevity of cut rose.

## Figures and Tables

**Figure 1 genes-13-01989-f001:**
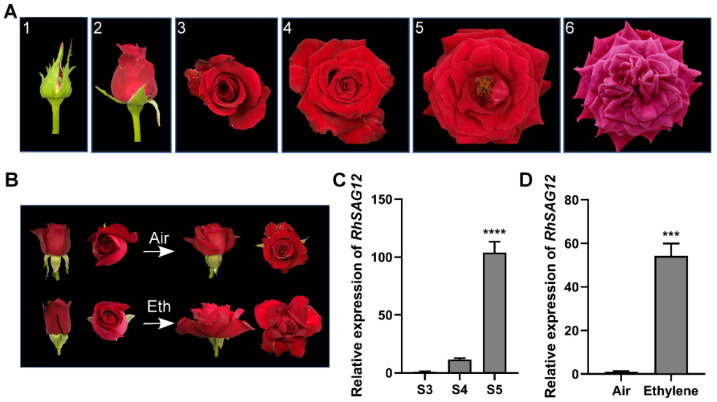
Effects of age and ethylene on rose phenotype. (**A**) Different flower opening stages of rose flowers. Six developmental stages of rose were divided into stage 1, pigmented bud with partially opened sepals; stage 2, bud with fully opened sepals; stage 3, flower with loose outermost petals; stage 4, flower with completely unfolded petals; stage 5, fully opened flower; and stage 6, flower with senesced petals. (**B**) Representative rose flowers after air or ethylene (Eth) treatment for 24 h. (**C**,**D**) Expression of *RhSAG12* during flower opening (**C**) and after ethylene treatment (**D**). *RhUBI2* was used as reference gene. Values are reported as mean ± SE (*n* ≥ 3). Student’s *t*-test, *** *p* < 0.001, **** *p* < 0.0001.).

**Figure 2 genes-13-01989-f002:**
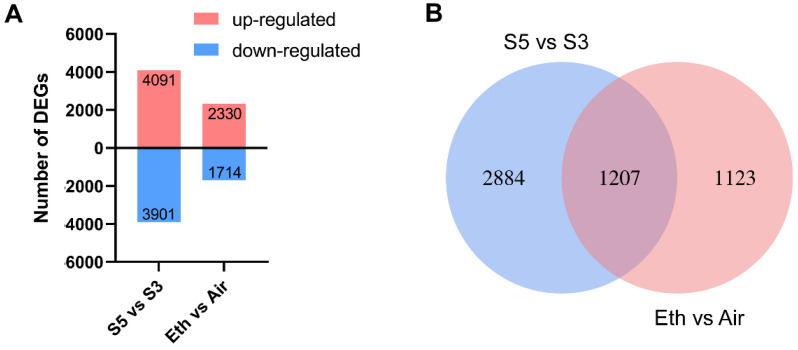
Comparison of development-dependent and ethylene-regulated genes in rose petals. (**A**) Numbers of differentially expressed genes (DEGs) at developmental stages and after ethylene treatment. DEGs were determined using fold changes ≥ 2 and *p*-value < 0.05. (**B**) Venn diagram of upregulated DEGs in groups representing stage 3 (S3), stage 5 (S5), and ethylene-treated petals (Eth).

**Figure 3 genes-13-01989-f003:**
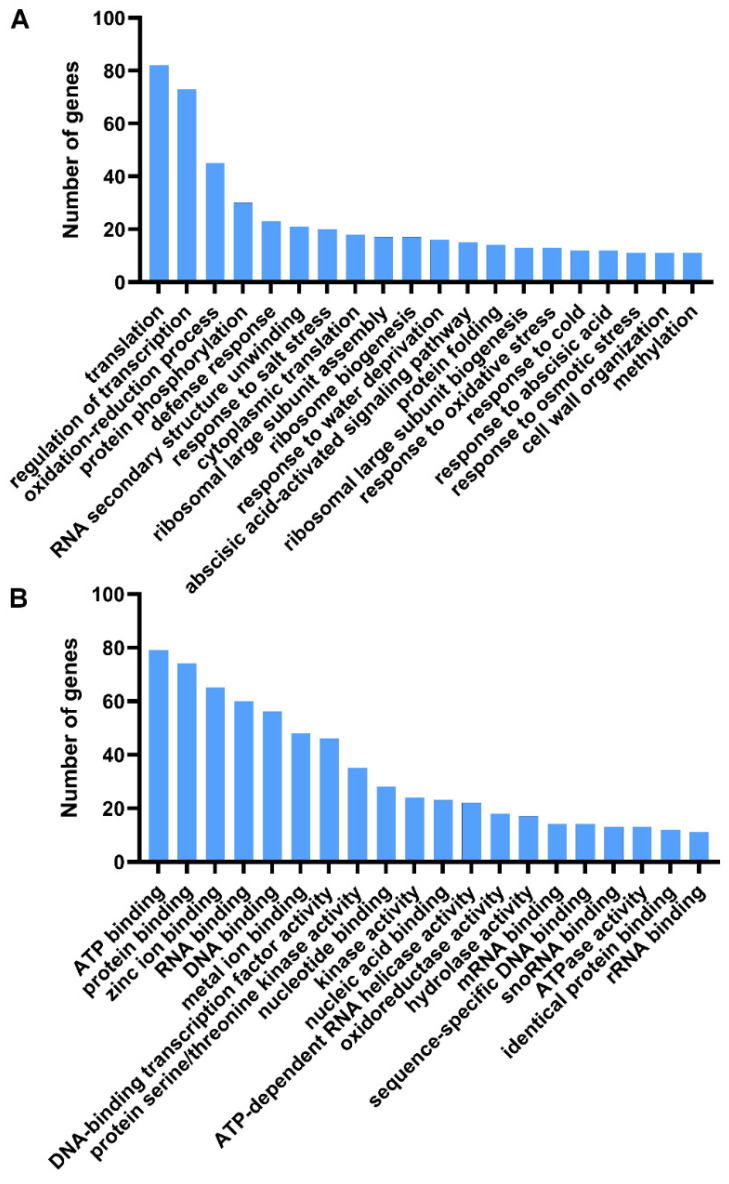
Gene ontology (GO) classification analysis of differentially expressed genes (DEGs). (**A**) GO analysis of DEGs assigned to biological processes. (**B**) GO analysis of DEGs assigned to molecular functions. Histograms illustrating number of DEGs assigned to different gene ontology terms.

**Figure 4 genes-13-01989-f004:**
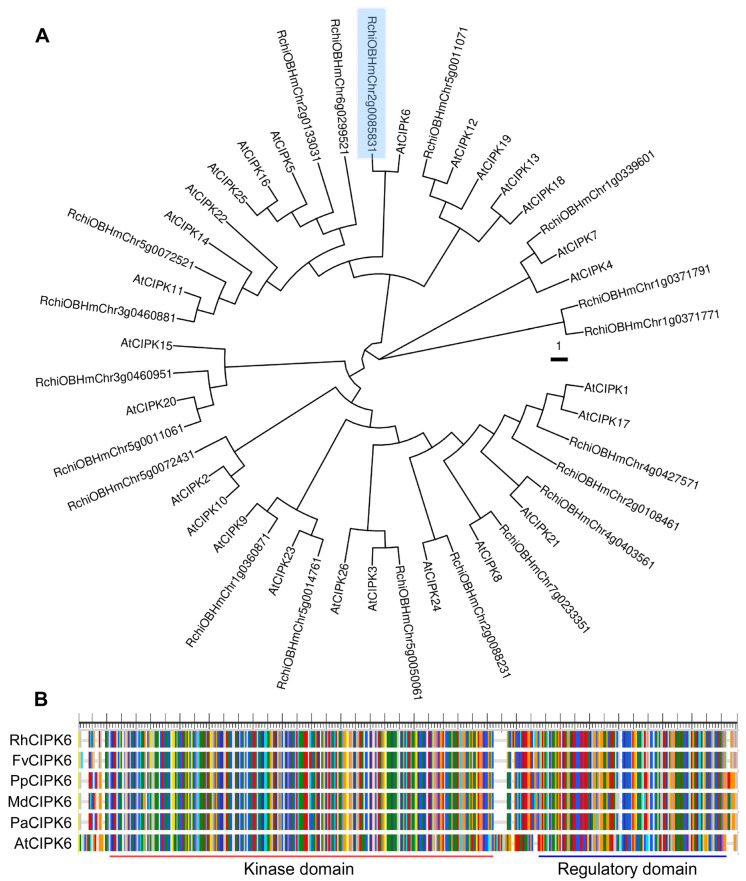
Phylogenetic analysis and sequence alignment of RhCIPK6. (**A**) Phylogenetic tree analysis of putative rose CIPK proteins with 26 CIPK proteins from *Arabidopsis*. (**B**) Multiple sequence alignment of CIPK6 proteins from different species. Conserved domains are represented below the alignment. At, *Arabidopsis thaliana*; Fv, *Fragaria vesca*; Pp, *Prunus persica*; Md, *Malus domestica*; Pa, *Prunus armeniaca*.

**Figure 5 genes-13-01989-f005:**
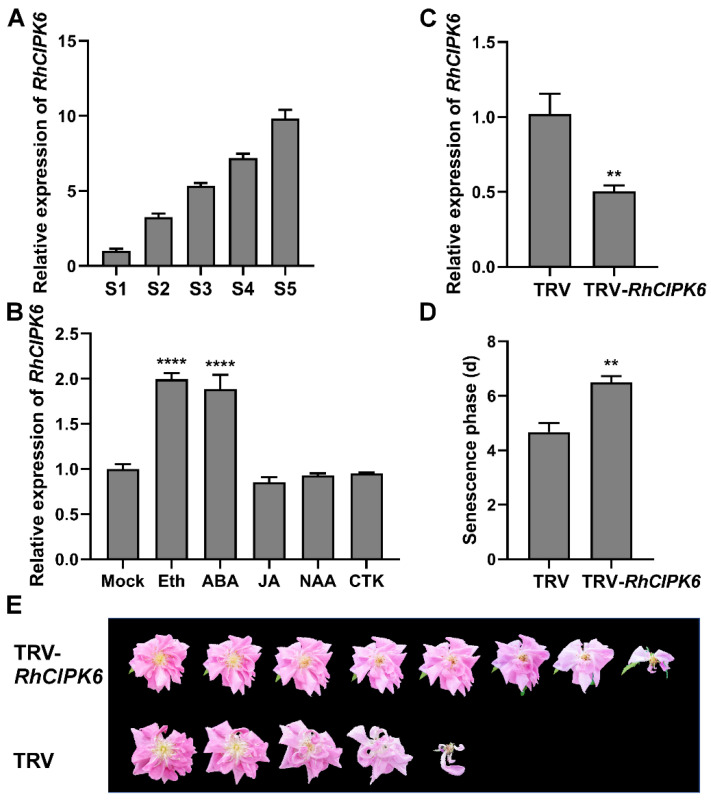
Effect of TRV-*RhCIPK6* on senescence of rose flowers. (**A**,**B**) Relative expression of *RhCIPK6* at different flower opening stages (**A**) and under different hormone treatments (**B**). Eth, 10 μL/L ethylene; ABA, 100 μM ABA; JA, 100 μM Me-JA; NAA, 100 μM NAA; CTK, 100 μM 6-BA. The flowers at stage 2 were used as material. Mock samples were treated with solvent only. (**C**) Relative *RhCIPK6* expression in petals determined in *RhCIPK6*-silenced and control plants by qRT-PCR. (**D**) Length of senescence phase in *RhCIPK6*-silenced and control plants. (**E**) Flower phenotype of *RhCIPK6*-silenced and TRV control plants. Phenotypes were recorded starting from the day with fully opened flower (the first day of stage 5). Experiments were independently performed three times with similar results. Values are reported as mean ± SE (*n* ≥ 3). Student’s *t*-test, ** *p* < 0.01, **** *p* < 0.0001.

**Figure 6 genes-13-01989-f006:**
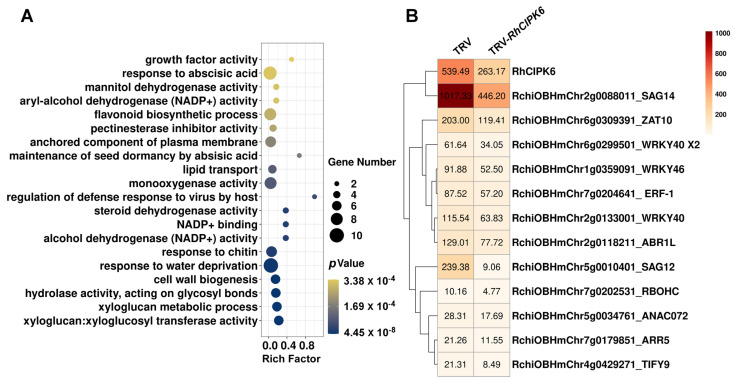
Analysis of differentially expressed genes in *RhCIPK6*-silenced petals. (**A**) GO biological process enrichment of DEGs in TRV-*RhCIPK6* petals compared with TRV control petals. Vertical axis indicates GO terms and abscissa represents the rich factor. The q value indicates significance of the rich factor. Size of dots indicates number of genes in the GO term. (**B**) Expression profile of screened genes identified in TRV-*RhCIPK6* petals compared with TRV control petals.

**Figure 7 genes-13-01989-f007:**
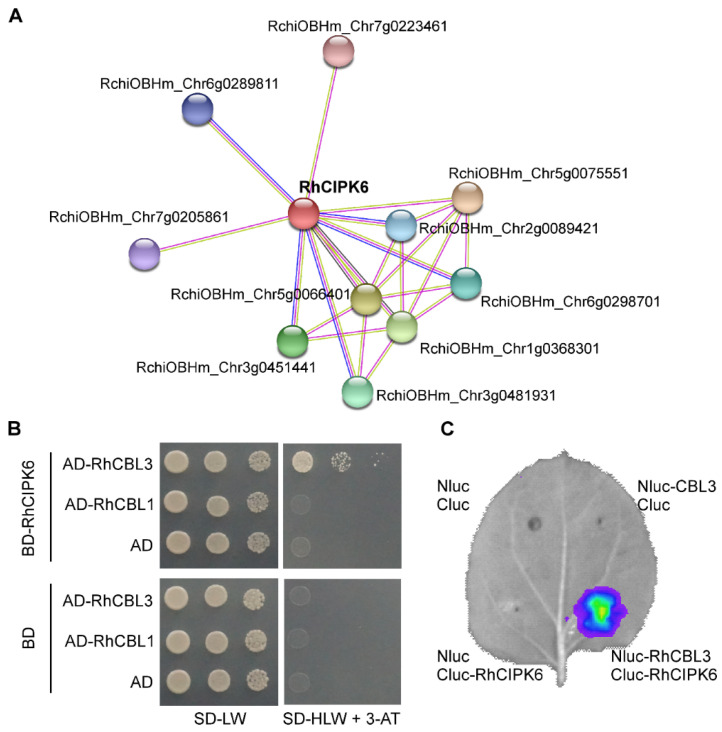
Identification of interacting protein of RhCIPK6. (**A**) Prediction RhCIPK6 protein–protein interaction network constructed using STRING database. PPI network of top 10 hub genes via information of STRING protein query with confidence score ≥ 0.7. Network nodes represent proteins. Edges indicate both functional and physical protein associations. (**B**) Results of yeast two-hybrid assay between RhCIPK6 and RhCBL1/3. CBL1/3 were fused to the GAL4-activating domain (AD) and RhCIPK6 was fused to the GAL4 DNA-binding domain (BD). (**C**) Interaction of RhCIPK6 and RhCBL3 as tested by the split-luciferase complementation assay in *N. benthamiana*. RhCIPK6 and RhCBL3 were constructed in cLUC and nLUC vectors, respectively. Empty constructs were used as negative controls.

## Data Availability

The data presented in this study are available in the article and Appendix A.

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
