# Peer review of "Protein Kinase RhCIPK6 Promotes Petal Senescence in Response to Ethylene in Rose (Rosa Hybrida)"

_genes, 2022, doi:10.3390/genes13111989_

Round 1

Reviewer 1 Report

This manuscript submitted by Yanqing et al. report the function of rose CIPK in petal senescence, and this study will be important. However, I found some of the authors’ explanations difficult to follow. Authors should review their manuscript and Figures in detail again and resubmit them.

Authors should describe the information of SAGs in Introduction. I could understand that CIPK6 and CBL3 bind, but how does this work on downstream genes to promote senescence? Authors could add that information to discussion.  

Line 19: Authors should confirm the numbers of TF and protein kinase. The numbers are described as 87 transcription factors and 49 protein kinases in line19, although these are described as 82 transcription factors and 48 protein kinases in line 195.  

Line 168: The total number of DEGs in stage5 versus stage 3 is described as 7,993 in line 168, but table S1 shows that the number is 7992. Please confirm.

Line 172: The numbers of DEGs in ethylene versus Air are different in line 172 and Figure A. In Figure A, the numbers are too few. Authors need to confirm and revise it. 

Line 173: The number of up-regulate genes by ethylene treatment is described 2,330 genes. However, the number is described 3,504 in pink area of Venn diagram in Figure 2B. Please confirm.

Line 176: The number of DEGs is described 1,207 as the number candidate genes, and the same number is describer in line 184 and 195. However, it is shown that the number is 1,257 in line 18 and Figure B and Table S1. Authors need to confirm and revise it.

L280: Why did authors focus on CBL1 and CBL3? Were only CBL1 and CBL3 detected in rose petal? 

L290 Figure 6B ---> Figure7B

L292 Figure 6C---> Figure7C

Figure 5E: Which stage of flowering is each of the roses shown in Figure 5E? Authors should add the information in caption or photo.

Author Response

Reviewer 1:

This manuscript submitted by Yanqing et al. report the function of rose CIPK in petal senescence, and this study will be important. However, I found some of the authors’ explanations difficult to follow. Authors should review their manuscript and Figures in detail again and resubmit them.

We greatly appreciate the detailed and thorough reviews, which allowed us to clarify, expand on and improve our manuscript. We have carefully revised the manuscript according to your comments and marked up the changes using the “Track Changes” function in our revised manuscript.

Comment 1: Authors should describe the information of SAGs in Introduction. I could understand that CIPK6 and CBL3 bind, but how does this work on downstream genes to promote senescence? Authors could add that information to discussion.

Response: Thanks for the valuable suggestion. We added relevant description of “SAGs” in the introduction section, and revised our discussion about “possible downstream during petal senescence” in our manuscript.

Comment 2: Line 19: Authors should confirm the numbers of TF and protein kinase. The numbers are described as 87 transcription factors and 49 protein kinases in line19, although these are described as 82 transcription factors and 48 protein kinases in line 195.

Response: Thanks for your valuable comment. Due to our carelessness, we did not carefully check the data in the abstract. We have changed the sentence from “87 transcription factors and 49 protein kinases” to “82 transcription factors and 48 protein kinases” in the abstract section of our revised manuscript.

Comment 3: Line 168: The total number of DEGs in stage5 versus stage 3 is described as 7,993 in line 168, but table S1 shows that the number is 7992. Please confirm.

Response: Thanks for your valuable comment. We have changed the number from “7993” to “7992” in our revised manuscript.

Comment 4: Line 172: The numbers of DEGs in ethylene versus Air are different in line 172 and Figure A. In Figure A, the numbers are too few. Authors need to confirm and revise it.

Response: Thanks for your valuable comment. Due to our carelessness, we did not carefully check the figure in the manuscript. We have changed Figure 2A in our revised version, and checked all the figures throughout the text.

Comment 5: Line 173: The number of up-regulate genes by ethylene treatment is described 2,330 genes. However, the number is described 3,504 in pink area of Venn diagram in Figure 2B. Please confirm.

Response: Thanks for your valuable comment. Due to our carelessness, we did not carefully check the figure in the manuscript. We have changed Figure 2B in our revised version, and checked all the figures throughout the text.

Comment 6: Line 176: The number of DEGs is described 1,207 as the number candidate genes, and the same number is described in line 184 and 195. However, it is shown that the number is 1,257 in line 18 and Figure B and Table S1. Authors need to confirm and revise it.

Response: Thanks for your valuable comment. Due to our carelessness, we did not carefully check all the data in the manuscript. We have changed the number from “1257” to “1207” in the text and Figure 2B and the genes in the Table S1 in our revised version.

Comment 7: L280: Why did authors focus on CBL1 and CBL3? Were only CBL1 and CBL3 detected in rose petal?

Response: We found four candidate CBL protein by using PPI network analysis, and only CBL1 and CBL3 were abundant in the petal. And there are other CBL proteins can be detected in rose petal, but we did not investigate their interaction with CIPK6 in this work.

Comment 8: L290 Figure 6B ---> Figure7B

Response: We have changed the figure citation from “6B” to “7B”, and checked all the citation throughout the text.

Comment 9: L292 Figure 6C---> Figure7C

Response: We have changed the figure citation from “6C” to “7C”, and checked all the citation throughout the text.

Comment 10: Figure 5E: Which stage of flowering is each of the roses shown in Figure 5E? Authors should add the information in caption or photo.

Response: According to the valuable comments, we added the stage information in the figure legend of Figure 5E.

Reviewer 2 Report

Intersting work about gene expression assoaciated to petal senescence in rose. However major revisions must be performed before publication:

In Material and Method sections, RNA-Seq analysis should be described before qPCR analysis. RNA-Seq data must be validated by uwsing qPCR.

RNA-Seq protocol and bioinformatic analysis must be completed and better explained.

Protein–protein interaction network analysis msut be better explained.

Legend of Figure 5 must be improved including description of the assayed treatments.

Format of Figure 6 must be improved incerasing letter size. 

Discussion section is very poor and must be completed. Results are very interesting and allow a better discussion.

A new section with the conclusions must be incorporated including main impications of the obtained results from a breeding and production point of view.

Author Response

Reviewer 2:

Intersting work about gene expression assoaciated to petal senescence in rose. However major revisions must be performed before publication:

We greatly appreciate careful review and constructive suggestions regarding our manuscript. We have considered these comments carefully and revised the manuscript in accordance with the comments and marked all the amends on our revised manuscript.

Comment 1: In Material and Method sections, RNA-Seq analysis should be described before qPCR analysis. RNA-Seq data must be validated by using qPCR.

Response: Thanks for your valuable comment. We have moved the qPCR analysis after the RNA-Seq analysis in the Materials and Methods section of our revised manuscript.

Comment 2: RNA-Seq protocol and bioinformatic analysis must be completed and better explained.

Response: Thanks for your valuable suggestion. We have added the detailed description of RNA-seq and bioinformatic analysis in the Materials and Methods section of our revised manuscript.

Comment 3: Protein–protein interaction network analysis must be better explained.

Response: Thanks for your valuable suggestion. We have added the description of Protein–protein interaction network analysis in the “Materials and Methods” section and “figure legend” of our revised manuscript.

Comment 4: Legend of Figure 5 must be improved including description of the assayed treatments.

Response: Thanks for your valuable comment. We have improved the description of assayed treatments in legend of Figure 5, and modified the figure in our revised manuscript.

Comment 5: Format of Figure 6 must be improved increasing letter size.

Response: Thanks for your valuable suggestion. We have changed Figure 6 with bigger letter size in our revised version.

Comment 6: Discussion section is very poor and must be completed. Results are very interesting and allow a better discussion.

Response: As suggest, we rewrote the discussion section in our revised version.

Comment 7: A new section with the conclusions must be incorporated including main implications of the obtained results from a breeding and production point of view.

Response: As suggest, we added the conclusion section in our revised version.

Round 2

Reviewer 1 Report

The sentence and figures I pointed out have been appropriately corrected.

Reviewer 2 Report

Authors have revised correctly the manuscript